# CONVOLUTION AND POOLING OPERATION MODULE WITH ADAPTIVE STRIDE PROCESSING EFFECT

## ABSTRACT

Convolutional neural network is one of the representative models of deep learning, which has a wide range of applications. Convolution and pooling are two key operations in convolutional neural networks. They play an important role in extracting input features and mapping low-level semantic features to high-level semantic features. Stride is an important parameter involved in convolution and pooling operations, which refers to the distance of each slide of the convolution kernel (pooling kernel) during the convolution (pooling) operation. The stride has an impact on the granularity of feature extraction and the selection (filtering) of features, thus affecting the performance of convolutional neural networks. At present, in the training of convolutional neural networks, the content of convolution kernel and pooling kernel can be determined by the optimization algorithm based on gradient descent. However, the stride usually cannot be treated similarly, and can only be selected manually as a hyperparameter. Most of the existing related works choose a fixed stride, for example, the value is 1. In fact, different tasks or inputs may require different stride for better model processing. Therefore, this paper views the role of stride in convolution and pooling operation from the perspective of sampling, and proposes a convolution and pooling operation module with adaptive stride processing effect. The feature of the proposed module is that the feature map finally obtained by convolution or pooling operation is no longer limited to equal interval downsampling (feature extraction) according to a fixed stride, but adaptively extracted according to the changes of input features. We apply the proposed module on many convolutional neural network models, including VGG, Alexnet and MobileNet for image classification, YOLOX-S for object detection, Unet for image segmentation, and so on. Simulation results show that the proposed module can effectively improve the performance of existing models.

## 1 INTRODUCTION

The research on convolutional neural networks started from 1980s to 1990s. Time delay network and LENET-5 were the earliest convolutional neural networks. After the 21st century, with the proposal of deep learning theory and the improvement of computing equipment, convolutional neural networks have developed rapidly and been applied in computer vision(Krizhevsky et al., 2012), natural language processing(Qiuqiang Kong, 2020) and other fields.

Operators in convolutional neural network include convolution operator and pooling operator. The elements of the convolution operator include the size of the convolution kernel, the numerical size of the convolution kernel, the stride of the convolution operation and so on. The elements of the pooling operator include stride, padding, and so on. Convolution neural network in the stride is to point to: convolution kernels or pooling operator acting on by convolution or by pooling area, convolution kernels or pooling operator each sliding distance, convolution and pooling operation is to extract the characteristics of the input and the lower sampling, stride for feature extraction of the characteristics of grain size and trade-off (filtering), which influence the properties of convolution neural network. In the current convolutional neural network, the stride of convolution or pooling operator convolution or pooling operation is manually selected as a hyperparameter and is fixed. The fixed stride means that the sliding distance of convolution kernel (pooling kernel) is the same

each time, and the fixed stride is 1, which will cause redundancy in extracting features. Different tasks may require different stride, for example, the pool operation stride is fixed to 2, and fixed to 2 is not fixed to 1, and may cause the loss of features. Moreover, the stride directly obtained by learning added to the model can not be backpropagated in training to update.

Self-adaptation is the process of automatically adjusting the processing method, processing sequence, processing parameters, boundary conditions or constraint conditions according to the data characteristics of the processed data in the process of processing and analysis, so that it can adapt to the statistical distribution characteristics and structural characteristics of the processed data, so as to achieve the best processing effect. The introduction of self-adaptation in convolutional neural network can make the network more adaptable to input changes, so as to make the whole network play a better effect, such as the improvement of processing speed, accuracy and recall. At present, adaptation in convolutional neural networks includes adaptation in determining the depth of convolutional operators(Veit & Belongie, 2019), adaptation in determining the width of convolutional operators(Jiaqi Ma, 2018), and adaptation in determining the parameters of convolutional neural networks (Adam W. Harley, 2017), Adaptation in determining the receptive field of convolution operator(Xizhou Zhu, 2019), pooling operator determination(Chen-Yu Lee, 2015), etc.

Since the stride obtained by direct learning cannot be backpropagated to update the stride when added to the model, it cannot be determined by the optimization algorithm based on gradient descent, and can only be selected manually as a hyperparameter. Therefore, the current academic research on the stride adaptation in convolutional neural networks is basically blank.(Rachid Riad, 2022) proposed an adaptive stride method for images in the frequency domain, and obtained the stride by learning method to filter out high-frequency signals in the frequency domain. However, the learning stride is different from the stride concept in traditional convolutional neural networks. In this paper, a convolution and pooling operation module with adaptive stride for traditional convolution operation, namely CAS(Convolution operation adaptive stride) and PAS(Pooling operation adaptive stride) module, is proposed.

We find that in the traditional convolution or pooling operator (operation), a fixed convolution or pooling stride S (S>1) The convolution or pooling operation results obtained can essentially be regarded as those obtained by adopting convolution or pooling operation results with stride 1 and then taking down-sampling at equal intervals (that is, discarding or suppressing part of the results). Based on the insight, further promotion, in order to obtain more general convolution with adaptive stride or pooling operation effect, can consider to use stride 1 convolution or pooling of input feature maps are operation, obtain the preliminary convolution operation or pooling results (initial feature maps), and then trying to based on the input features to generate a suitable mask characteristic figure, The mask feature map is combined with the initial feature map, and some results are discarded or suppressed (for example, insignificant results are set to 0), so as to obtain the final feature map with adaptive stride convolution or pooling operation effect. The most important aspect of adaptation is that the feature map finally obtained by convolution or pooling operation is no longer limited to equal interval downsampling or feature extraction according to a fixed stride S, but adaptively extracted according to the change of input features.

## 2 METHODS

In convolutional neural networks, convolution operation or pooling operation can extract features and map the underlying semantic features to higher-level semantic features and the role of down-sampling. Classification needs local features and localization needs global features. In convolutional neural network, the convolution or pooling operation stride is fixed as 1, which can extract detailed features, but will cause feature redundancy. Because the position of the object in each picture is changed, the stride sliding distance is the same each time but not fixed to 1 may cause the loss of features. Therefore, it is the correct way to use the stride in convolution or pooling operation to adaptively determine the stride for each movement process of convolution kernel and pooling operator according to the input feature map. It is hoped that convolution or pooling operation can focus on the key areas needed.

This is slightly different from the traditional concept of fixed stride, which means that the convolution kernel or pooling operator moves the same distance every time. As shown in Figure 1, the red region is the convolved or pooled region.

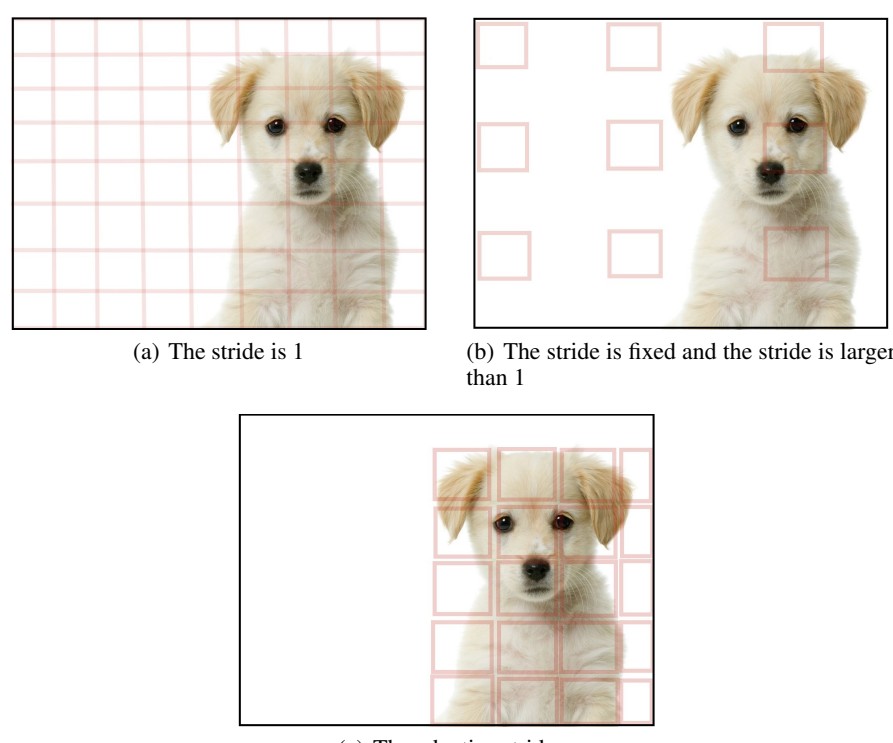

(a) The stride is 1

(b) The stride is fixed and the stride is larger than 1

(c) The adaptive stride

Figure 1: Convolution or pooling effect with different stride

## 2.1 CAS MODULE

### 2.1.1 RELATIONSHIP BETWEEN THE SAMPLING

The stride of the convolution operation refers to the distance that the convolution kernel slides each convolution. Figure 2,3 illustrates the convolution operation with stride 1 and stride 2 and compares the results.

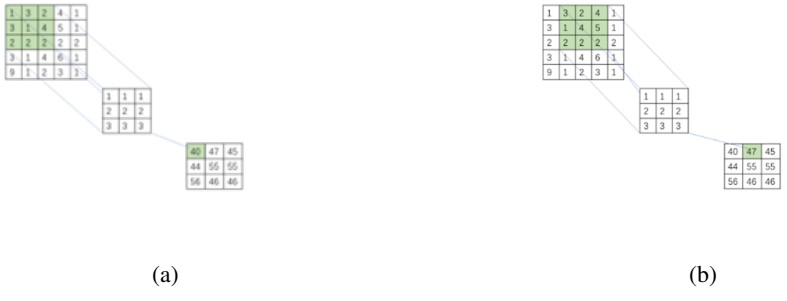

(a)                                              (b)

Figure 2: Stride is 1

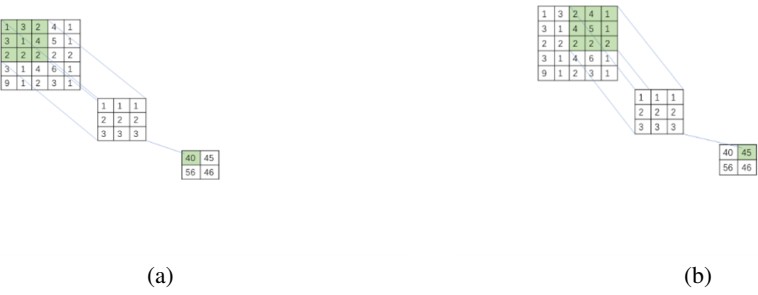

(a)                                                                                    (b)

Figure 3: Stride is 2

Figure 4 shows the comparison of convolution results with stride 1 and stride 2. The left is the result with stride 1, the right is the result with stride 2, and the middle shows the relationship between them.

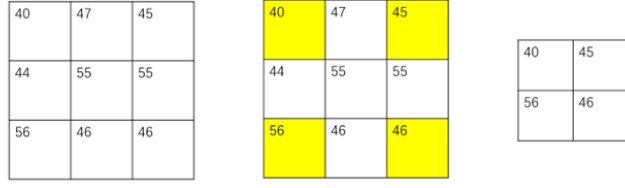

Figure 4: Comprision bwteen different stride result

It can be seen that the fixed convolution stride S (S>1), the convolution operation results obtained can be essentially regarded as those obtained by taking equal interval downsampling on the basis of convolution operation results with stride 1.

### 2.1.2 MASK FEATURE MAP

In order to obtain the result of convolution operation with stride not 1, we can sample the result feature map of convolution operation with stride 1. In order to realize sampling, we introduce the concept of mask feature map. The size of the mask feature map is the same as that of the convolution operation with stride 1. There are some zeros and some positive numbers in the mask feature map, and the positions of the non-zero positive numbers are equally spaced. Then, the mask feature map and the result feature map of convolution operation with stride 1 are used for the bitwise multiplication processing, as shown in Figure 5.

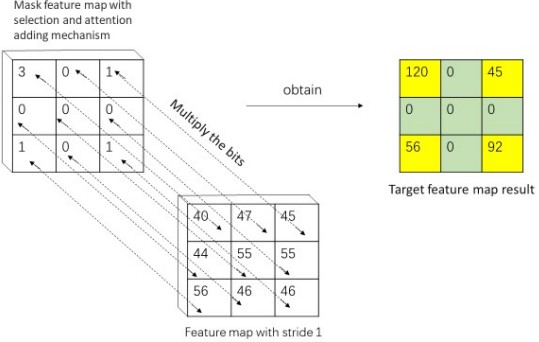

Figure 5: Through the mask feature map sampling process

The mask feature map is processed by bitwise multiplication with the resulting feature map of convolution with stride 1. The position of the mask 0 corresponds to the volume of stride 1 and the position of the feature map of the operation result. Set this parameter to 0. The position selection corresponding to the positive number is retained, and the attention mechanism is added. The value at the position of 0 will not continue to propagate forward and can be regarded as blank, which is similar to the size change of the feature map resulting from the convolution operation whose stride is not 1 in the traditional concept.

### 2.1.3 ADAPTIVE STRIDE

As shown in Figure 6, the positions of non-zero positive numbers in the mask feature map are not equally spaced. The adaptive stride method proposed in this paper is no longer limited to fixed convolution stride S (S>1), but adaptively sampling the convolution operation result with stride 1 according to the input feature map, which can make the convolution operation focus more on the important information of the feature map. The position of 0 in the target feature map is regarded as the convolution operation stride of this position plus one, and the position of the color box is the actual convolution area, as shown in Figure 6.

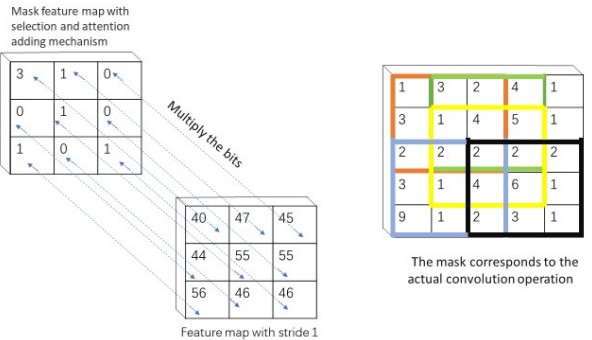

Figure 6: Through the mask feature map sampling process

### 2.1.4 ADAPTIVITY GENERATED BY THE MASK

The generation of the mask feature map is obtained through the three-layer convolutional layer convolution processing of the feature map with the result of step 1 in the convolutional neural network. In addition, the mask feature map is processed with activation transformation (such as Relu, etc.)

to add selection effect and attention mechanism effect to the mask feature map. The reason for selecting the three-layer convolutional layer is to generate the feature map with the same size as the feature map resulting from the convolution operation with step stride of 1. At the same time, the three-layer convolutional layer is regarded as a complex function(Jifeng Dai, 2017), and the mask feature map is generated adaptively according to the input feature map. The reason for choosing Relu is that the Relu function can set the position required by the mask feature map to 0, and can carry out back propagation in the training.

### 2.1.5 THE CAS OVERALL METHOD

The overall method of CAS is as follows: The input feature map is convolved by convolution layer with fixed stride, and the initial feature map is generated. The input feature map is convolved by three convolutional layers, and the mask feature map is generated. The mask feature map was processed with activation transformation (such as Relu(Glorot X, 2010), etc.), and selection action and attention mechanism were added to the mask feature map. The mask feature map is multiplied by the initial feature map to generate the target feature map, and the convolution operation result with adaptive stride effect is obtained, as shown in Figure 7.

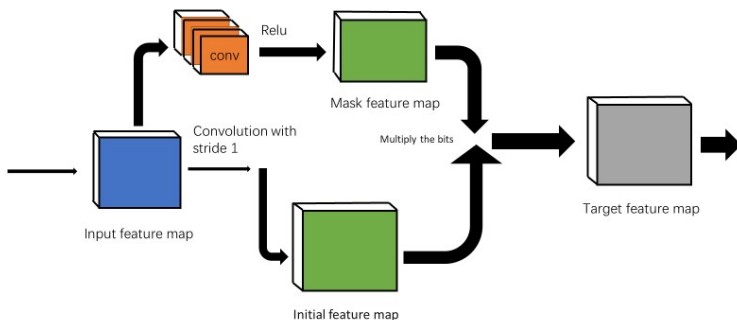

Figure 7: Through the mask feature map sampling process

### 2.2 PAS MODULE

The function of pooling layer is feature extraction and downsampling, and its stride is usually determined by cross validation. The overall method of PAS is as follows: The input feature map is pooled by pooling layer with fixed stride, and the initial feature map is generated. The input feature map is convolved by three convolutional layers, and the mask feature map is generated. The mask feature map was processed with activation transformation (such as Relu(Glorot X, 2010), etc.), and the selection action and attention mechanism were added to the mask feature map. The target feature map is generated by the bitwise multiplication of the mask feature map and the initial feature map, and the pooling result with the effect of adaptive stide is obtained.

## 3 EXPERIMENT

We carried out the convolution operation in the classification network, object detection network, segmentation network adaptive stride experiments. The adaptive stride experiment of pooling operation on classification network is also carried out. The classification accuracy has been improved, and visual experiments have been done to verify the effectiveness and generality of our method.

Through experiments, we find that the adaptive stride plays a role in extracting local features. Classification task is the need for local characteristics, the more the farther the back-end network characteristic figure network own extract local features, but there is interference and limited data set, the partial feature of the network itself extraction is not so perfect, so our CAS module should be

added in the network of the middle and front, it can better help the network to extract local features. Moreover, we found that CAS module can be added to the middle and front of the object detection model network or segmentation network, because the localization needs relatively global features, and the adaptive stride helps extract local features to improve both classification and localization. It can also be added to the front of the classification decoupled branch of the object detection model with decoupled classification localization task, because the decoupled stem layer is a global feature.

Classification network we selected classic classification networks VGG, Alexnet and MobileNet, all of which added CAS module in the middle and front of the network, Alexnet is shown in Figure 7. The accuracy has been improved, as shown in Table 1. For object detection network, YOLOX-S network with decoupling of classification and positioning is selected, and CAS module is added to the stem layer with decoupling of classification and positioning, as shown in Figure 8. The accuracy has been improved, as shown in Table 2. We selected Unet for the segmentation network, and added CAS module to the subsampling convolution layer at the second layer of the network. Both accuracy and mIU have been improved, indicating that CAS module is helpful for classification and positioning, as shown in Table 3. We selected VGG and Alexnet for PAS module, and the model accuracy was improved, as shown in Table 4.

### 3.1 EXPERIMENT ON ADAPTIVE SRIDE OF CONVOLUTION OPERATION

#### 3.1.1 CLASSIFICATION OF NETWORK

Because of the overlap of Trick effect, we chose the most original version of VGG network and Alexnet network,the effect is shown in Table 1.

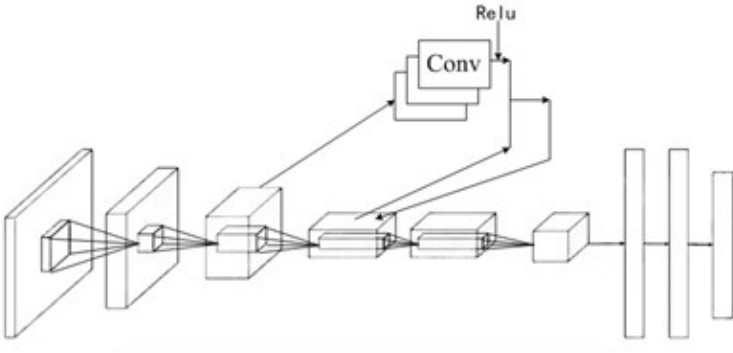

Figure 8: Alexnet adds adaptive stride

Table 1: Classification of network

| Model | Dataset | Acc | CAS module Acc |
|---|---|---|---|
| VGG | Cifar10 | 55.11% | 55.83% |
| Alexnet | Cifar10 | 68% | 70% |
| MobileNet | Cifar10 | 90.680% | 91.840% |

Through the experiment, it is found that adding CAS module to the classification network can help the network to better extract local features and improve the classification accuracy.

#### 3.1.2 OBJECT DETECTION

Localization needs global features while classification needs local features. The YoloX-S model decouples the classification task from the localization task. Because the stem layer is used for both classification and localization, it is a global feature. Our module is applicable to the classification network and the front-end of the network, so on the YOLOX-S model, our CAS module is added

to the stem layer and the first layer of the classification branch connection. This is shown in Figure 9,and the effect is shown in Table2.

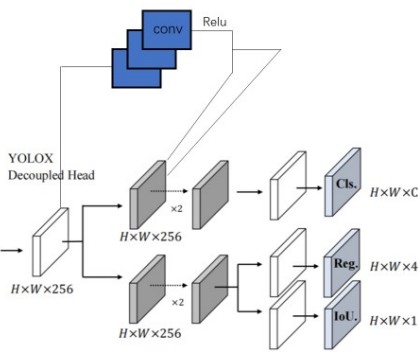

Figure 9: YOLOX with CAS

Table 2: YOLOX-S

| Dataset | condition | YOLOX-S AP | CAS module AP |
|---------|-----------|------------|---------------|
| COCO | Iou=0.50;0.95 area=all maxDets=100 | 40.4% | 41.0% |
| COCO | Iou=0.50 area=all maxDets=100 | 59.2% | 59.6% |
| COCO | Iou=0.75 area=all maxDets=100 | 43.6% | 44.5% |
| COCO | Iou=0.50;0.95 area=small maxDets=100 | 23.2% | 22.6% |
| COCO | Iou=0.50;0.95 area=medium maxDets=100 | 44.5% | 45.3% |
| COCO | Iou=0.50;0.95 area=largel maxDets=100 | 53.1% | 54.3% |

### 3.1.3 SEGMENTATION NETWORK

Positioning is a relatively global feature, and our CAS module is also helpful for the extraction of relatively global features, so we add the CAS module in the middle and front of Unet. It was found that both classfication and positioning were improved,as is shown in Table 3.

Table 3: Unet

| | Model | Dataset | Acc | Acc cls | Mean iu | fwavacc |
|---|-------|---------|-----|---------|---------|---------|
| original | Unet | Voc | 74.40% | 14.70% | 10.75% | 58.71% |
| CAS | Unet | Voc | 75.02% | 18.74% | 13.99% | 59.58% |

### 3.2 EXPERIMENT OF ADAPTIVE STRIDE OF POOLING OPERATION

In the experiment of the adaptive step size of pooling operation, we chose the original VGG, Alexnet network and Alexnet network with many tricks added, and added the PAS module in the second pooling layer. The original step size of pooling layer was selected as 2. The improvement obtained is shown in Table 4, which verifies the effectiveness of the adaptive stride on the pooling layer.

Table 4: PAS experiment

| Model | Dataset | Acc | PAS |
|---|---|---|---|
| VGG | Cifar10 | 55.11% | 56.23% |
| Alexnet | Cifar10 | 68% | 71% |
| Alexnet with tricks | Cifar10 | 74.140% | 77.583% |

### 3.3 VISUALIZATION EXPERIMENT

In the visualization experiment, we output the original image and mask for comparison, as shown in Figure 10. The left is the original image and the right is the mask. The experimental results show that the position of non-zero positive number in the mask is exactly the position of the object, which verifies our idea of extracting local features with adaptive stride.

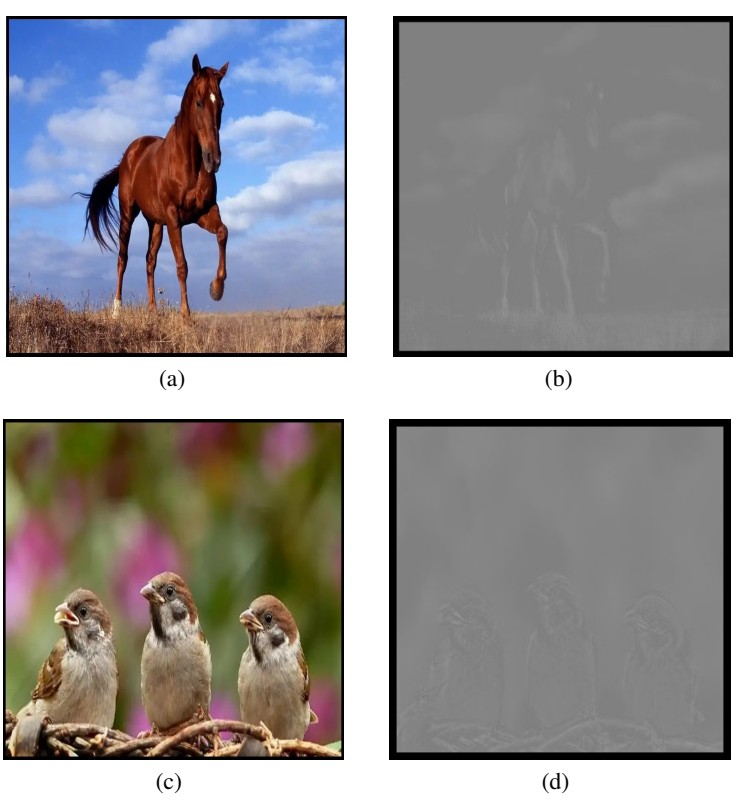

(a)                    (b)

(c)                    (d)

Figure 10: Visualization experiment

## 4 CONCLUSION AND FUTURE WORK

We design an adaptive stride module in convolutional neural network, which verifies the effectiveness and generality of our method in many models. The adaptive stride module does play a role in extracting local features, making the convolution operation or pooling operation more focused on the required features. The effect of the adaptive stride is derived from the mask feature map and the result feature map with stride 1 is selected. The mask is generated by three convolutional layers and processed by activation transformation. In the future work, we will explore more mask generation methods and find other modules with adaptive stride.

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
