# OpenReview forum: "$$CONVOLUTION AND POOLING OPERATION MODULE WITH ADAPTIVE STRIDE PROCESSING EFFEC$$"
_ICLR.cc/2023/Conference — Submitted to ICLR 2023_

### Official Review · Reviewer_3ok7 · 2022-10-23

**Confidence:** 3
**Correctness:** 4
**Technical Novelty And Significance:** 2
**Empirical Novelty And Significance:** Not applicable
**Recommendation:** 5

**Clarity, Quality, Novelty And Reproducibility:**


The authors could improve the clarity and the quality of the papers.
Many spelling errors can be found in the paper and also sections of the paper are difficult to understand.

- Spelling errors: Figure 4: Comprision bwteen different stride result ( CompArison, bEtween ....)

On Figure 4, the authors should add more details on the caption such that the reader understand that the two figures at the left represent the result with stride=1 and the figure at the right the result with stride = 2

Figure 5:
Could you explain how you obtain the mask of Figure 5 ?
It seems that the bitwise multiplication gives ((120, 0, 45), (0, 0, 0), (56, 0, 46)). Could you clarify this please ?

**Strength And Weaknesses:**

The topic of this paper is very interesting and the main strength relies on its technical novelty, since the authors propose a module based on a dynamic stride for convolution operation. However, this paper lacks mathematical models that can help to understand difference between fixed and dynamic strides.

When using a dynamic stride, it is not clear if the size of the model (number of parameters will change) or the speed (number of operations). Could you discuss also this question ?

**Summary Of The Paper:**

While a lot of CNN architectures are based on a fixed stride, this paper analyzes convolutional operations with an adaptitive stride. They developed a module that extract feature map by convolution with a stride that changes adaptively according to the changes of input features.

They apply the proposed module on many convolutional neural network models such as VGG, Alexnet, MobileNet, YOLOX-S and UNET. Simulation results are also provided to show the efficiency of the module.


**Summary Of The Review:**

This paper raises the question of adaptive stride in convolutional neural network. To my knowledge, this is something new and. Even if the topic of this paper is very interesting, the authors still have a lot of work to do, particularly on the mathematical modeling of the problem and in the writing of the paper.

---

### Official Review · Reviewer_9Nmu · 2022-10-24

**Confidence:** 5
**Clarity, Quality, Novelty And Reproducibility:** No novelty.
**Correctness:** 1
**Technical Novelty And Significance:** 1
**Empirical Novelty And Significance:** Not applicable
**Recommendation:** 1

**Strength And Weaknesses:**

If this paper is generated by AI then it is really a good job. If this paper is written by a real human, then it is its weakness. The paper is filled with trivial facts and no innovation.

The modules in the paper seem to be trivial special cases of off-the-shelf tools such as the deformable convolution. The experiments are almost nothing.

**Summary Of The Paper:**

"We find that in the traditional convolution or pooling operator (operation), a fixed convolution or
pooling stride S (S>1) The convolution or pooling operation results obtained can essentially be
regarded as those obtained by adopting convolution or pooling operation results with stride 1 and
then taking down-sampling at equal intervals (that is, discarding or suppressing part of the results)."

**Summary Of The Review:**

If this paper is generated by AI then it is really a good job. If this paper is written by a real human, then it is its weakness. The paper is filled with trivial facts and no innovation.

The modules in the paper seem to be trivial special cases of off-the-shelf tools such as the deformable convolution. The experiments are almost nothing.

---

### Official Review · Reviewer_DpD2 · 2022-10-26

**Confidence:** 4
**Correctness:** 2
**Technical Novelty And Significance:** 1
**Empirical Novelty And Significance:** 1
**Recommendation:** 1

**Clarity, Quality, Novelty And Reproducibility:**

it's very hard to find something positive to say about the paper in its current state.
Clarity is very bad (c.f. comments on writing above), novelty is impossible to evaluate since the model description is very vague and experiments are weak. The lack of details also makes the method unreproducible at the moment.

**Strength And Weaknesses:**

Strengths:
 * Learning strides by backpropagation is still an open question as previous work that address this task suffer from a significant increase in computation cost compared to standard strided convolutions.

Weaknesses:
* The paper needs a complete rewriting, see points below:
  * The paper is very hard to read, with unfinished sentences, useless sentences (eg. the first paragraph of the introduction should just be removed) and many grammar and syntax errors. An example is the first paragraph of the Methods section which I just do not understand.
  * The total absence of notations and model details only provides the reader with a vague and abstract idea of the method.
  * Figures 2 and 3 take almost a page just to illustrate the basic mechanism of striding, which is a waste of space.
  * Many typos such as "comprision" in Figure 4, "Classification of network" as a title Section 3.1.1.
  * Paragraph 2.2 is just a rewriting of paragraph 2.1.5, with copy-pasted sentences (!)
* The method in itself is very questionable. First, unless I missed it, the method seems to only mask the target feature map leaving zero entries, which means that regardless of the "adaptive stride" learned by the model, there is no actual striding applied and the spatial dimensions remain constant in the entire network. This is completely different from striding, which effectively downsamples an image such that kernels of upper layers process inputs at lower resolutions. Moreover, the method suffers from the same limitation of [DiffStride](https://arxiv.org/abs/2202.01653)  as it first needs to compute an unstrided convolution + a mask produced itself by a convnet, which is overall much more costly than a simple strided convolution. This observation coupled to the fact that the model does not actually downsample but rather keep the spatial dimensions constant along the network lets me think that this model is much slower than baselines.
* Not comparing to DiffStride and claiming that "the current academic research on the stride adaptation [...] is basically blank", the absence of confidence intervals, the lack of details on the experimental setting and the absence of time/space complexity analysis makes it impossible to evaluate the method in the current state.

**Summary Of The Paper:**

Authors introduce a method to learn strides by backpropagation.

**Summary Of The Review:**

This paper needs to be rewritten entirely and better experiments realized (more details, more baselines) to convince the reader that the method is worth. I also would appreciate that authors explain how downsampling is actually performed, because otherwise the title and claims are misleading.

---

### Decision · Program_Chairs · 2023-01-20

**Decision:**

Reject

**Justification For Why Not Higher Score:**

- unclear writing to the degree that it is difficult to understand what exactly the method does
- missing related work (for instance, DiffConv https://arxiv.org/abs/2202.01653, deformable convolutions https://arxiv.org/abs/1703.06211)
- related to this, missing experimental comparisons to related work - both in terms of performance and computational cost
Given all this, it is impossible to tell if the method is indeed new and useful

**Justification For Why Not Lower Score:**

N/A

**Metareview: Summary, Strengths And Weaknesses:**

The paper proposes a variant of convolution and poling operations with adjustable strides. The module is tested with a few ConvNet architectures and vision tasks.

The reviewers agree that the paper is not fit for publication. The main cons are
- unclear writing to the degree that it is difficult to understand what exactly the method does
- missing related work (for instance, DiffConv https://arxiv.org/abs/2202.01653, deformable convolutions https://arxiv.org/abs/1703.06211)
- related to this, missing experimental comparisons to related work - both in terms of performance and computational cost

Given all this, it is impossible to tell if the method is indeed new and useful, so I recommend rejection at this point.